# Chitosan as an Adjuvant to Enhance the Control Efficacy of Low-Dosage Pyraclostrobin against Powdery Mildew of *Rosa roxburghii* and Improve Its Photosynthesis, Yield, and Quality

**DOI:** 10.3390/biom12091304

**Published:** 2022-09-16

**Authors:** Cheng Zhang, Qinju Li, Jiaohong Li, Yue Su, Xiaomao Wu

**Affiliations:** 1Guizhou Food Quality and Safety Technology Service Platform, School of Public Health, Guizhou Medical University, Guiyang 550025, China; 2Institute of Crop Protection, College of Agriculture, Guizhou University, Guiyang 550025, China; 3Department of Food and Medicine, Guizhou Vocational College of Agriculture, Qingzhen 551400, China

**Keywords:** powdery mildew, pyraclostrobin, chitosan, *Rosa roxburghii*, reducing application of chemical fungicides

## Abstract

Powdery mildew is the most serious fungal disease of *Rosa roxburghii* in Guizhou Province, China. In this study, the control role of chitosan-assisted pyraclostrobin against powdery mildew of *R**. roxburghii* and its influences on the resistance, photosynthesis, yield, quality and amino acids of *R**. roxburghii* were evaluated. The results indicate that the foliar application of 30% pyraclostrobin suspension concentrate (SC) 100 mg L^−^^1^ + chitosan 500 mg L^−^^1^ displayed a superior control potential against powdery mildew, with a control efficacy of 89.30% and 94.58% after 7 d and 14 d of spraying, respectively, which significantly (*p* < 0.01) exceeded those of 30% pyraclostrobin SC 150 mg L^−^^1^, 30% pyraclostrobin SC 100 mg L^−^^1^, and chitosan 500 mg L^−^^1^. Simultaneously, their co-application could effectively enhance their effect on the resistance and photosynthesis of *R. roxburghii* leaves compared to their application alone. Meanwhile, their co-application could also more effectively enhance the yield, quality, and amino acids of *R. roxburghii* fruits compared to their application alone. This work highlights that chitosan can be applied as an effective adjuvant to promote the efficacy of low-dosage pyraclostrobin against powdery mildew in *R. roxburghii* and improve its resistance, photosynthesis, yield, quality, and amino acids.

## 1. Introduction

*Rosa roxburghii* Tratt., a promising natural medicine and third-generation fruit rich in vitamin C, flavonoids, superoxide dismutase (SOD), and minerals, has various beneficial functions, such as enhancing immunity, decreasing blood pressure, and regulating the digestive system, as well as anti-cancer, anti-oxidation, and anti-radiation effects, etc. [1,2,3,4,5,6]. Recently, as an agricultural industry boosting affluence and revitalizing rural areas, the *R. roxburghii* industry has flourished rapidly in Guizhou Province in China, with a planting areas of over 170,000 hm^2^, representing the largest production area in the world [3,6] However, powdery mildew caused by *Sphaerotheca* sp. constantly occurs and seriously restricts the growth, yield, and quality of *R. roxburghii*, as well as frequently generating 30~40% economic losses [1,7]. As *R. roxburghii* is mainly produced in China, its powdery mildew also occurs frequently in the Sichuan, Yunnan, Chongqing, southwestern Shaanxi, Hubei and Hunan production areas in China, often causing equally great economic losses [7,8]. Consequently, some chemical fungicides and natural products for controlling powdery mildew have been proposed by local scholars. For example, Yan et al. [9] found that 10% benazoxystrobin suspension concentrate (SC) could effectively control powdery mildew with a control efficacy of 83.59%~ 94.33%. Moreover, Yan et al. [10] also reported that 6% ascorbic acid aqueous solutions induced the resistance of *R. roxburghii* to powdery mildew with a control efficacy of 61.45%. Subsequently, Li et al. [7] reported that 1.0~1.5% chitosan could also induce *R**. roxburghii* resistance to powdery mildew, with an induced control effect of 69.30%~72.87%. Nonetheless, considering the serious harmfulness of powdery mildew, it is of major significance to excogitate various alternative and practicable control strategies for the healthy development of the *R**. roxburghii* industry.

Pyraclostrobin, an efficient broad-spectrum strobilurin fungicide containing a pyrazole structure, is one of the most widely used and sold fungicides in the world [11,12]. Due to its good protective, therapeutic and systemic properties, pyraclostrobin is widely registered for controlling many plant diseases of fruit trees, vegetables and grains caused by various fungal pathogens, including *Ascomycetes*, *Basidiomycetes*, *Hemimycetes*, and *Oomycetes*, etc. [13,14]. It mainly inhibits the electron transfer between cytochrome b and c1 in mitochondrial respiration of pathogenic cells, so that mitochondria cannot normally provide the required energy for cell metabolism, thereby achieving a bactericidal effect [15,16,17]. In a previous report from our research group, Wu et al. [18] found that 30% pyraclostrobin SC could effectively control powdery mildew with a control efficacy of more than 90%, and notably promote its yield and quality. Nevertheless, with the increasing of its application areas, targets, and dosage, its potential risk to the environment, animals, and humans is also increased [19,20,21,22]. In the meantime, it easily triggers pathogen resistance with the growth of application frequency [23,24]. Noteworthily, Wang et al. [25] reported that oligosaccharins could be applied as an adjuvant to enhance tebuconazole’s control of soft rot disease of kiwifruit and reduce tebuconazole application. In that way, whether some natural products or biomolecules can be applied as an adjuvant to pyraclostrobin to more effectively control powdery mildew of *R**. roxburghii*, reduce pyraclostrobin application and mitigate potential risks should be further explored and studied.

Chitosan, a natural biomolecule widely applied in the food, agriculture, medicine and cosmetic fields, has various outstanding advantages, such as nontoxicity, renewability, biocompatibility, etc. [26,27,28]. In agriculture, it can be used as a growth enhancer for promoting a plant’s growth and also as a bio-fungicide and inductor for managing diseases and pests [29,30,31]. The putative mechanisms of chitosan-mediated plant growth regulation are shown in Figure 1. Effectively, Li et al. [7] found that chitosan could induce *R**. roxburghii* to resist powdery mildew and promote its photosynthesis and quality. Subsequently, the authors further demonstrated that chitosan had good toxicity against *Sphaerotheca* sp. with an EC_50_ value of 416.21 mg kg^−1^, and the co-application of allicin and chitosan effectively controlled powdery mildew with an efficacy of 85.97%, as well as reliably enhancing the resistance, growth and quality of *R. roxburghii* [32]. Meanwhile, Wang et al. [33] found that chitosan could be used as an adjuvant to improve the efficacy of isopyrazam·azoxystrobin against leaf spot disease of kiwifruit and decrease its application dosage. In our early report, we also demonstrated that chitosan was also a good adjuvant of the bio-fungicide tetramycin for controlling leaf spot disease of kiwifruit and improving its resistance, photosynthesis, and quality [34]. Accordingly, whether chitosan can enhance the control efficacy of pyraclostrobin against powdery mildew of *R. roxburghii* and reduce pyraclostrobin application is worth further study. 

In the present study, the control efficacy of pyraclostrobin + chitosan against powdery mildew in *R. roxburghii* was investigated for the first time. Simultaneously, the effects of pyraclostrobin + chitosan on the resistance and photosynthetic capacities of *R. roxburghii* leaves were determined. Subsequently, the effects of pyraclostrobin + chitosan on the yield, quality, and amino acids of *R. roxburghii* fruits were also evaluated. This work presents a novel, feasible and alternative approach toward using natural product-assisted chemical fungicides in controlling the powdery mildew of *R. roxburghii* and reducing chemical fungicide application.

## 2. Materials and Methods

### 2.1. Fungicide, Atomizer and Chemical

In this study, 30% pyraclostrobin SC was provided by Zhongke Green Biological Engineering Co. Ltd. (Jinan, China). Chitosan (deacetylation ≥ 90.00%) was produced by Mingrui Bioengineering Co. Ltd. (Zhenzhou, China). The electrostatic atomizer was produced by Qiming Machinery Co. Ltd. (Taizhou, China), and its nominal capacity, spraying flow rate and working pressure were 16 L, ≤2.7 L min^−1^ and 0.15~0.40 MPa, respectively. All chemicals were of chromatographic or analytical grade. 

### 2.2. Field Orchard

An orchard of *R. roxburghii* in Longli country, Guizhou province, China (26°54′36′′ N, 106°95′13′′ E) was used in the field experiment. The planted cultivar was high-quality ‘Guinong 5′, which is widely planted in Guizhou province. The tree age of the *R. roxburghii* plants was eight years old, and the planting density was 106 plants per 666.7 m^2^. Additionally, the altitude, temperature, annual sunshine, annual rainfall, and frostless season in *R. roxburghii* orchard were approximately 1384 m, 13.9 °C, 1265 h, 1100 mm and 280 d, respectively. The fertility information for *R. roxburghii* orchard soils is displayed in Table 1, while its pH value and exchangeable calcium were 6.37 and 17.06 cmol kg ^−1^, respectively.

### 2.3. Field Control Experiment of Powdery Mildew 

The foliar spray method was applied for the control experiment of powdery mildew in *R**. roxburghii*. Meanwhile, a completely randomized experimental method was used for the delineation of the experimental plots. Five treatments were designed for controlling powdery mildew: (1) 30% pyraclostrobin SC 100 mg L^−^^1^ + chitosan 500 mg L^−^^1^ (P 100 + C 500), (2) 30% pyraclostrobin SC 150 mg L^−^^1^ (P 150), (3) 30% pyraclostrobin SC 100 mg L^−^^1^ (P 100), (4) chitosan 500 mg L^−^^1^ (C 500), and (5) clear water (control). For the preparation of P 100 + C 500, the appropriate amount of water was firstly used to dissolve 30% pyraclostrobin SC and chitosan, respectively, then the 30% pyraclostrobin SC and chitosan diluents were mixed, and the required water was finally complemented. Each treatment consisted of three replicates, and each plot contained nine trees, and the five trees on the diagonal were applied for testing. Considering that the young leaves, young stems, flower buds, flowers, and young fruits of *R. roxburghii* are the main organs damaged by powdery mildew, the leaves, stems, flowers and buds of each *R. roxburghii* plant (including) were sprayed with 1.50 L of fungicide liquid on April 3 and April 10 in 2021.

The control efficacy of *R. roxburghii* powdery mildew was investigated on April 17 and April 24 in 2021 as described by Li et al. [7,32]. The classification of the incidence degree was as follows: 0 degree was no incidence, 1 degree was 1~2 diseased lobules with thin hyphae, 2 degree was 3~4 diseased lobules with thick hyphae, 3 degree was 5~6 diseased lobules with dense hyphae, and 4 degree was more than 7 diseased lobules with dense hyphae. The disease index and control effect of powdery mildew in *R. roxburghii* were calculated using Equations (1) and (2), respectively.
Disease index = 100 × ∑ (Disease degree value × Leaf number within each degree)/(Total leaf number × the highest degree)(1)
Control effect (%) = 100 × (1 − Disease index of fungicide/Disease index of control)(2)

### 2.4. Determination of Disease Resistance and Photosynthesis Parameters of R. roxburghii

*R. roxburghii* leaves from the middle, east, west, south and north orientations of each tree tested were randomly collected on April 24 in 2021. A mixture of collected leaves in each plot were used for determining the disease resistance parameters of *R. roxburghii* leaves, including total phenolics, total flavonoids, soluble protein, soluble sugar, proline (Pro), malonaldehyde (MDA), SOD activity and polyphenoloxidase (PPO) activity, as described by Wang et al. [35] and Zhang et al. [36,37]. In the meantime, the photosynthesis parameters of *R. roxburghii* leaves including chlorophyll content, photosynthetic rate (Pn), transpiration rate (Tr) and water use efficiency (WUE) were also measured on April 24 in 2021 according to Zhang et al. [34]. Chlorophyll content was determined by an UV-5800PC spectrophotometer at 663 nm and 645 nm with acetone–ethanol (*v*/*v*, 2:1) extraction. Additionally, a portable LI-6400XT photosynthesis measurement system (LI-COR Inc., Lincoln, NE, USA) was applied for monitoring the Pn and Tr of *R. roxburghii* leaves at 8:00–10:00 a.m. on April 24 in 2021.

### 2.5. Determination of Yield, Quality and Amino Acids of R. roxburghii

*R. roxburghii* fruits from the middle, east, west, south and north orientations of each tree tested were randomly collected on September 3 in 2021. The single fruit weight and yield per plant of *R. roxburghii* were measured as described by Li et al. [7,32]. Meanwhile, the quality parameters of *R. roxburghii* fruits including vitamin C, soluble solid, soluble protein, soluble sugar, total acidity, total flavonoids, total triterpenes, and SOD were also determined as described by Wang et al. [35] and Zhang et al. [36,37]. Subsequently, a HPLC system (ThermoFisher U3000, Thermo Fisher Scientific (China), Shanghai, China) was used for determining 17 hydrolyzed amino acids of *R. roxburghii* fruits according to Zhang et al. [34]. Additionally, essential amino acids (EAA), nonessential amino acids (NAA), total amino acids (TAA), the percentage of EAA in TAA, and EAA/NAA were then calculated according to the contents of 17 hydrolyzed amino acids. 

### 2.6. Statistical Analyses

The mean value ± standard deviation (SD) of three replicates was displayed. SPSS 18.0 software (SPSS Inc., Chicago, IL, USA) was used to check the significant differences of data, and a one-way analysis of variance (ANOVA) was applied. Figures were drawn using Origin 10.0 software (OriginLab, Northampton, MA, USA).

## 3. Results

### 3.1. Field Control Effect of Pyraclostrobin and Chitosan against Powdery Mildew

The control effects of pyraclostrobin + chitosan, pyraclostrobin and chitosan against powdery mildew of *R. roxburghii* are depicted in Table 2. After 7 d and 14 d of spraying with fungicides, P 100 + C 500, P 150, P 100, and C 500 significantly (*p* < 0.01) attenuated the disease index of powdery mildew, and P 100 + C 500 exhibited an optimal effect. P 100 + C 500 displayed a superior control potential for powdery mildew, with control effects of 89.30% and 94.58% after 7 d and 14 d of spraying, respectively, which significantly (*p* < 0.01) exceeded those of P 150, P 100, and C 500. The control effects against powdery mildew were ranked as follows: P 100 + C 500 > P 150 > P 100 > C 500, which all showed preferable persistence. Although C 500 displayed a relatively inferior control potential against powdery mildew, its control effects still reached 58.73% and 73.87% after 7 d and 14 d of spraying, respectively. Meanwhile, the pyraclostrobin amount in P 100 + C 500 was effectively reduced compared with P 150. These results show that chitosan induced a favorable control effect on powdery mildew and could significantly enhance the control effect of a low dosage of pyraclostrobin against powdery mildew and effectively decrease the application amount of pyraclostrobin. 

### 3.2. Influence of Pyraclostrobin and Chitosan on Disease Resistance of R. roxburghii Leaves

The influence of pyraclostrobin and chitosan on the total phenolics, total flavonoids, soluble protein, and soluble sugar of *R. roxburghii* leaves is depicted in Figure 2. Compared with the control, P 100 + C 500, P 150, and C 500 significantly (*p* < 0.05) augmented the total phenolics, total flavonoids, soluble protein, and soluble sugar contents of *R**. roxburghii* leaves, while P 100 also significantly (*p* < 0.05) enhance their total flavonoids and soluble protein contents. Moreover, the total flavonoids, soluble protein, and soluble sugar contents of *R. roxburghii* leaves treated with P 100 + C 500 significantly (*p* < 0.05) exceeded those of P 150, P 100, and C 500. Additionally, the total phenolics content of *R. roxburghii* leaves treated with P 100 + C 500 significantly (*p* < 0.05) exceeded that of P 150 and P 100, as well as demonstrating no significant (*p* < 0.05) differences with that of C 500. Simultaneously, the phenolics, flavonoids, soluble protein, and soluble sugar contents were not significantly (*p* < 0.05) different in P 150 and C 500, but those of C 500 were slightly higher than those of P 150. These results show that as compared to the application of pyraclostrobin or chitosan alone, pyraclostrobin + chitosan could effectively improve the total phenolics, total flavonoids, soluble protein, and soluble sugar contents of *R. roxburghii* leaves, thereby promoting the resistance of *R. roxburghii* to powdery mildew.

The influence of pyraclostrobin and chitosan on the Pro, MDA, SOD and PPO activities of *R. roxburghii* leaves is shown in Figure 3. Compared with the control, P 100 + C 500, P 150, and C 500 significantly (*p* < 0.05) increased the Pro content, SOD and PPO activities of *R**. roxburghii* leaves and reduced their MDA content, while P 100 could only significantly (*p* < 0.05) promote their SOD activity and reduce their MDA content. Furthermore, the Pro content, SOD and PPO activities of *R. roxburghii* leaves treated with P 100 + C 500 significantly (*p* < 0.05) exceeded those of P 150, P 100, and C 500, and their MDA content was significantly (*p* < 0.05) lower than that of P 150, P 100, and C 500. Meanwhile, the Pro content, MDA content, SOD and PPO activities of *R. roxburghii* leaves also showed no significant (*p* < 0.05) differences in the P 150 and C 500 treatments, but those of C 500 were slightly higher than those of P 150. These results further indicate that chitosan used together with pyraclostrobin could effectively enhance the Pro content, SOD and PPO activities of *R. roxburghii* leaves and reduce their MDA content, reliably improving the resistance of *R**. roxburghii* to powdery mildew.

### 3.3. Influence of Pyraclostrobin and Chitosan on Photosynthetic Capacity of R. roxburghii Leaves

The influence of pyraclostrobin and chitosan on the chlorophyll, Pn, Tr, and WUE of *R. roxburghii* leaves is displayed in Figure 4. Compared with the control, P 100 + C 500, P 150, and C 500 significantly (*p* < 0.05) enhanced the chlorophyll, Pn, and Tr of *R. roxburghii* leaves, while P 100 also significantly (*p* < 0.05) improved their Pn and Tr. Moreover, the chlorophyll, Pn, and Tr of *R**. roxburghii* leaves treated with P 100 + C 500 were 6.25 μg g^−^^1^, 8.32 μmol CO_2_ m^−2^ s^−1^, and 2.91 mmol H_2_O m^−2^ s^−1^, respectively, being 1.06-, 1.17-, 1.07-, and 1.22-fold, 1.11-, 1.21-, 1.10-, and 1.36-fold, and 1.10-, 1.18-, 1.09-, and 1.30-fold higher compared to those of P 150, P 100, C 500 and the control, respectively. Simultaneously, the chlorophyll, Pn, and Tr of *R. roxburghii* leaves treated with C 500 were slightly higher than those of P 150, but there were no significant differences. Nevertheless, the water use efficiency of *R. roxburghii* leaves was not significantly (*p* < 0.05) different in the four treatments. The results presented here demonstrate that the co-application of pyraclostrobin and chitosan could more effectively enhance the chlorophyll, Pn, and Tr of *R. roxburghii* compared to their application alone, thereby promoting its favorable growth.

### 3.4. Influence of Pyraclostrobin and Chitosan on Yield and Quality of R. roxburghii

The influence of pyraclostrobin and chitosan on weight and yield per plant of *R**. roxburghii* fruits is displayed in Figure 5. Compared with the control, P 100 + C 500, P 150, P 100, and C 500 significantly (*p* < 0.05) enhanced the weight and yield of *R. roxburghii* fruits. P 100 + C 500 exhibited a superior increasing yield performance for *R. roxburghii* with a single fruit weight and fruit yield of 20.46 g and 7.18 kg per plant, respectively, being significantly (*p* < 0.05) higher by 1.12-, 1.25-, 1.09-, and 1.35-fold and 1.11-, 1.24-, 1.15-, and 1.55-fold compared to P 150, P 100, C 500 and the control, respectively. In addition, the weight and yield of *R. roxburghii* fruits were not significantly (*p* < 0.05) different in the P 150 and C 500 treatments. The results show that pyraclostrobin + chitosan could more effectively enhance *R. roxburghii* yield compared to pyraclostrobin or chitosan alone.

The influence of pyraclostrobin and chitosan on *R. roxburghii* quality is shown in Table 3. Compared with the control, P 100 + C 500, P 150, and C 500 could significantly (*p* < 0.05) enhance vitamin C, soluble solid, soluble protein, soluble sugar, total acidity, total flavonoids, total triterpenes, and SOD activity of *R. roxburghii* fruits. Vitamin C, soluble protein, and total triterpenes content of *R. roxburghii* fruits treated with P 100 + C 500 were significantly (*p* < 0.05) higher than those of P 150 and C 500, and their total flavonoids content was significantly (*p* < 0.05) higher than that of P 150. Simultaneously, compared with P 100, P 100 + C 500 could also significantly (*p* < 0.05) enhance the vitamin C, soluble solid, soluble protein, soluble sugar, total acidity, total flavonoids, total triterpenes, and SOD activity of *R. roxburghii* fruits. However, P 100 could only significantly (*p* < 0.05) improve their vitamin C, total acidity, total triterpenes, and SOD activity compared to control. These findings show that chitosan effectively improved the effect of pyraclostrobin on the nutritional quality of *R. roxburghii* fruits.

### 3.5. Influence of Pyraclostrobin and Chitosan on Amino Acids of R. roxburghii

The influence of pyraclostrobin and chitosan on amino acids of *R. roxburghii* fruits is depicted in Table 4. Compared with P 100 or the control, P 100 + C 500 could significantly (*p* < 0.05) increase the EAA, NAA, and TAA of *R. roxburghii* fruits, as well as raise the percentage of EAA in TAA and EAA/NAA. EAA and the percentage of EAA in TAA of *R. roxburghii* fruits treated with P 100 + C 500 significantly (*p* < 0.05) exceeded those of the P 150 or C 500 treatments. Additionally, the NAA, TAA and EAA/NAA of *R. roxburghii* fruits were not significantly (*p* < 0.05) different in P 100 + C 500, P 150, and C 500 treatments, while those of P 100 + C 500 were slightly higher than those of P 150 and C 500. These results reveal that the enhancing effect for amino acids of *R. roxburghii* fruits by pyraclostrobin + chitosan was superior to that of their application alone.

## 4. Discussion

Pyraclostrobin is a good protective, therapeutic and systemic fungicide that can inhibit the electron transfer between cytochrome b and c1 in the mitochondrial respiration of various fungal pathogens such as *Ascomycetes*, *Basidiomycetes*, *Hemimycetes*, and *Oomycetes*, etc. [13,14]. Chitosan has favorable antifungal activity and induces resistance to various plant diseases [28,29,30,31]. In the previous reports from our research group, Wu et al. [18] demonstrated that 30% pyraclostrobin SC 150 mg L^−^^1^ could effectively control powdery mildew in *R**. roxburghii* with a control efficacy of 91.01% after 14 d of spraying, and Li et al. [7] found that 1.0~1.5% chitosan could induce *R**. roxburghii*’s resistance to powdery mildew with a control effect of 69.30%~72.87% after 30 d of spraying. In this study, 30% pyraclostrobin SC 100 mg L^−^^1^ + chitosan 500 mg L^−^^1^ displayed a superior control potential for powdery mildew with control effects of 89.30% and 94.58% after 7 d and 14 d of spraying, respectively, which were significantly (*p* < 0.01) higher than the 81.77% and 88.76% of 30% pyraclostrobin SC 150 mg L^−^^1^, 68.47% and 80.39% of 30% pyraclostrobin SC 100 mg L^−^^1^, and 58.73% and 73.87% of chitosan 500 mg L^−^^1^, respectively. These results suggest that chitosan significantly enhanced pyraclostrobin’s control effect against powdery mildew and effectively decreased the application amount of pyraclostrobin. There is a notably synergetic effect: pyraclostrobin prevented pathogen infection and killed pathogens, while chitosan can both kill pathogens and induce a plant’s disease resistance.

Phenolics and flavonoids are precursors of lignin biosynthesis in plants, which can enhance lignification of host cells and thereby produce disease resistance [38]. Proteins are the metabolic basis of energy and material in plants, and Pro and soluble sugar participate in the regulation of cell permeability, while MDA is a product of membrane lipid peroxidation [38,39]. Meanwhile, SOD is a key protective enzyme for obliterating free radicals in plants, and PPO can catalyze the formation of lignin, phenolic oxidation products and quinones to produce disease resistance [38]. Many reports have also verified that chitosan can promote the increase in disease resistance substances in plants and stimulate the activity of defensive enzymes [28,29,30,31,32,33,34,39]. In previous reports from our research group, it was also shown that chitosan could effectively enhance the Pro, soluble sugar and flavonoid contents, SOD and POD activities of *R. roxburghii* leaves, and decrease their MDA [7,32]. In this study, pyraclostrobin + chitosan could effectively improve the total phenolics, total flavonoids, soluble protein, soluble sugar, Pro, SOD activity, PPO activity and MDA of *R. roxburghii* leaves, thereby more reliably enhancing the resistance of *R**. roxburghii* to powdery mildew. These results are consistent with the above reports, and also emphasize that chitosan and pyraclostrobin have a notably synergetic effect in enhancing the disease resistance of *R. roxburghii*.

Chlorophyll is a photosynthetic pigment, and photosynthesis is a physiological basis for the yield and quality of *R**. roxburghii*. Moreover, the main driving force for absorbing and transporting water and nutrients in plants is transpiration. Chakraborty et al. [27] demonstrated that chitosan can promote the photosynthetic rate of plants by increasing chlorophyll content, and further enhance their growth and development. The previous results in our research group also show that chitosan or chitosan + allicin effectively improved the chlorophyll and photosynthetic rate of *R. roxburghii* leaves [7,32]. In this study, the co-application of pyraclostrobin and chitosan could more effectively enhance the chlorophyll, photosynthetic rate, and transpiration rate of *R. roxburghii* compared to their application alone, thereby further promoting its favorable growth. Furthermore, the co-application of pyraclostrobin and chitosan could more effectively enhance the yield of *R. roxburghii* compared to their application alone. Chitosan can also activate the signal transduction and gene expression of auxin and cytokinin in plants, and further promote their growth and biomass formation [27,40]. This high yield of *R. roxburghii* treated with pyraclostrobin + chitosan derives from the synergetic contribution of the protection of pyraclostrobin and chitosan, as well as the growth promotion of chitosan.

Powdery mildew seriously restricts the quality of *R. roxburghii*. The favorable quality of *R. roxburghii* fruits depends on its good growth and no damage from disease or insect pests. In this study, under the premise that the co-application of pyraclostrobin and chitosan could effectively control powdery mildew in *R. roxburghii* and promote its growth, it was observed that they could also significantly (*p* < 0.05) promote the vitamin C, soluble solids, soluble protein, soluble sugar, total acidity, total flavonoids, total triterpenes, and SOD activity of *R. roxburghii* fruits. Meanwhile, the co-application of pyraclostrobin and chitosan could more effectively enhance the above quality parameters of *R. roxburghii* fruits compared to their application alone. Simultaneously, the co-application of pyraclostrobin and chitosan could also more effective improve the EAA, NAA, and TAA of *R. roxburghii* fruits than pyraclostrobin or chitosan alone. Based on the amino acid model proposed by WHO and FAO, the percentage of EAA in TAA and EAA/NAA of the superior quality foods are 40% and ≥0.6, respectively [41]. In this study, the percentage of EAA in TAA and EAA/NAA of *R. roxburghii* fruits treated with pyraclostrobin + chitosan were 19.17% and 0.26, which are closer to the ideal amino acid model values compared to pyraclostrobin, chitosan and the control. These results emphasize that chitosan is an effective tool with low-dosage pyraclostrobin for improving the quality and amino acids of *R. roxburghii* fruits.

Recently, growing attention has been focused on reducing the application of chemical fungicides and complementary or alternative measures for plant disease management [33,42]. Some reports demonstrated that chitosan can be an efficient adjuvant of isopyrazam azoxystrobin, allicin, and tetramycin against plant disease [32,33,34]. In the present study, the co-application of pyraclostrobin and chitosan could more efficiently control powdery mildew in *R**. roxburghii* and enhance its resistance, photosynthesis, yield, quality and amino acids compared with their application alone, as well as effectively reducing pyraclostrobin application. Meanwhile, chitosan is a nontoxic natural biomolecule, and the pyraclostrobin concentration in the combination of pyraclostrobin and chitosan is relatively low (100 mg L^−^^1^ or 10,000-fold diluted liquid). Furthermore, the safe interval time (126 days, 10 April 10 to 3 September) for *R. roxburghii* fruits was very long. Hence, the potential risks caused by pyraclostrobin and chitosan were almost nonexistent. This work emphasizes that 30% pyraclostrobin SC 100 mg L^−^^1^ + chitosan 500 mg L^−^^1^ can be recommended as a novel and feasible formula combination for managing powdery mildew in *R. roxburghii* and reducing chemical fungicide application.

## 5. Conclusions

In conclusion, chitosan effectively helped pyraclostrobin to control powdery mildew in *R. roxburghii* and decreased pyraclostrobin’s application. The co-application of pyraclostrobin and chitosan notably improved the total phenolics, total flavonoids, soluble protein, soluble sugar, and Pro contents and SOD and PPO activities in *R. roxburghii* leaves and decreased their MDA content, and reliably enhanced their chlorophyll contents, photosynthetic rate, and transpiration rate. Meanwhile, pyraclostrobin + chitosan exhibited a better effect in enhancing the yield, quality, and amino acids of *R. roxburghii* fruits compared to their application alone. This study highlights that chitosan together with pyraclostrobin can be proposed as a novel, green and feasible formula combination for controlling powdery mildew of *R. roxburghii* and reducing chemical fungicide application.

## Figures and Tables

**Figure 1 biomolecules-12-01304-f001:**
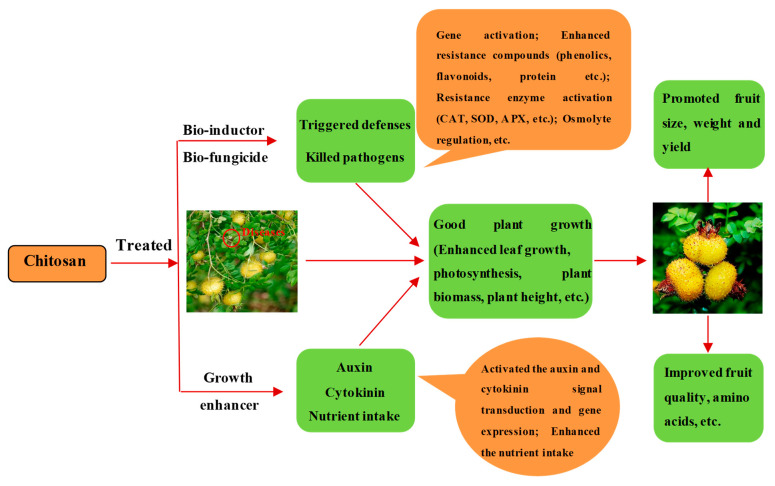
The putative mechanisms of chitosan-mediated plant growth regulation.

**Figure 2 biomolecules-12-01304-f002:**
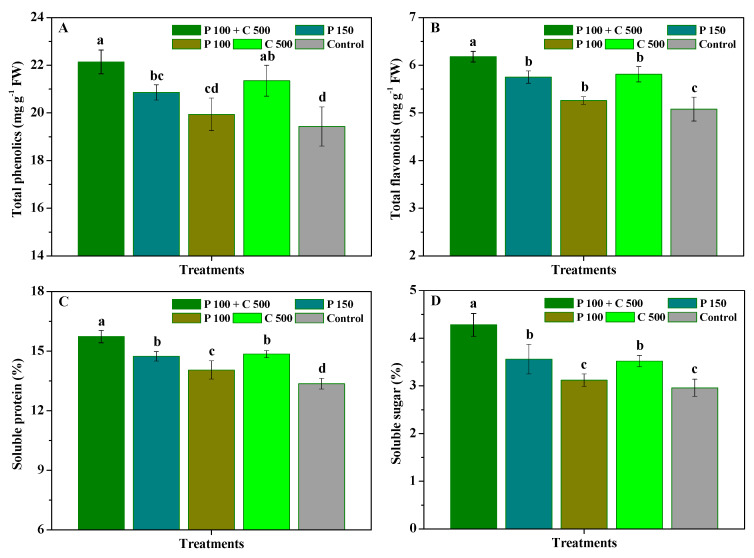
The influence of pyraclostrobin and chitosan on the total phenolics (**A**), total flavonoids (**B**), soluble protein (**C**), and soluble sugar (**D**) contents of *R. roxburghii* leaves. Error bars indicate the SD of the three replicates. Small letters represent significant differences on a 5% (*p* < 0.05) level.

**Figure 3 biomolecules-12-01304-f003:**
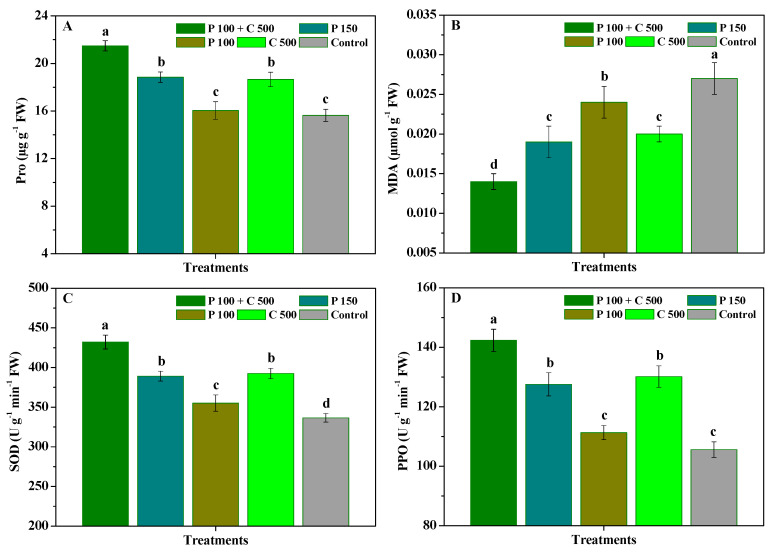
The influence of pyraclostrobin and chitosan on the Pro (**A**), MDA (**B**), SOD activity (**C**), and PPO activity (**D**) of *R**. roxburghii* leaves. Error bars indicate the SD of the three replicates. Small letters represent significant differences on a 5% (*p* < 0.05) level.

**Figure 4 biomolecules-12-01304-f004:**
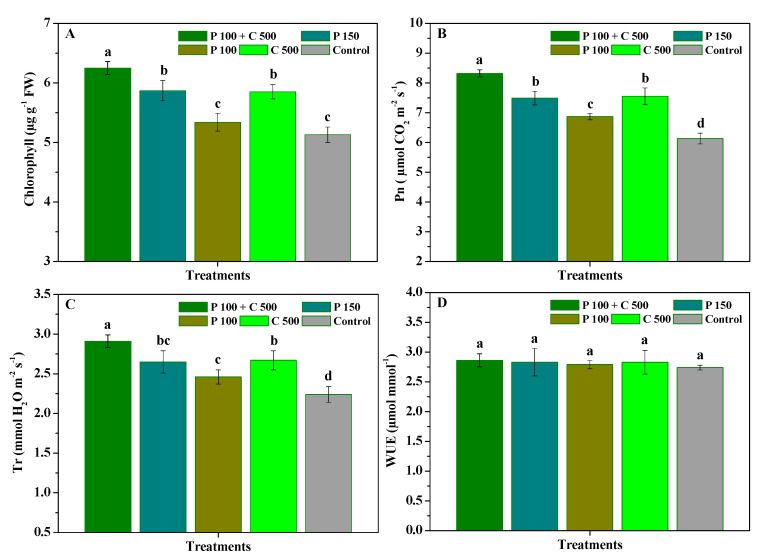
The influence of pyraclostrobin and chitosan on the chlorophyll (**A**), Pn (**B**), Tr (**C**) and WUE (**D**) of *R. roxburghii* leaves. Error bars indicate the SD of the three replicates. Small letters represent significant differences on a 5% (*p* < 0.05) level.

**Figure 5 biomolecules-12-01304-f005:**
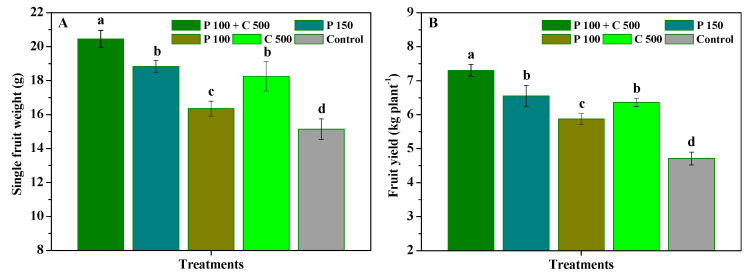
The influence of pyraclostrobin and chitosan on single fruit weight (**A**) and fruit yield per plant (**B**) of *R**. roxburghii*. Error bars indicate the SD of the three replicates. Small letters represent significant differences on a 5% (*p* < 0.05) level.

**Table 1 biomolecules-12-01304-t001:** The fertility information of soils in *R. roxburghii* orchard.

Indices	Content (g kg^−1^)	Indices	Content (mg kg^−1^)	Indices	Content (mg kg^−1^)
Organic matter	13.56	Available nitrogen	56.84	Available iron	6.59
Total nitrogen	1.42	Available phosphorus	4.66	Available boron	0.15
Total phosphorus	1.68	Available potassium	27.51	Exchangeable magnesium	308.37
Total potassium	1.24	Available zinc	0.71	Available manganese	15.46

**Table 2 biomolecules-12-01304-t002:** The control effect of pyraclostrobin and chitosan against powdery mildew.

Treatments	After 7 d of Spraying		After 14 d of Spraying	
	Disease Index	Control Effect (%)	Disease Index	Control Effect (%)
P 100 + C 500	1.06 ± 0.20 ^eE^	89.30 ± 1.50 ^aA^	0.82 ± 0.22 ^eE^	94.58 ± 1.51 ^aA^
P 150	1.79 ± 0.18 ^dD^	81.77 ± 2.85 ^bB^	1.70 ± 0.26 ^dD^	88.76 ± 1.77 ^bB^
P 100	3.10 ± 0.11 ^cC^	68.47 ± 2.56 ^cC^	2.97 ± 0.19 ^cC^	80.39 ± 1.16 ^cC^
C 500	4.05 ± 0.12 ^bB^	58.73 ± 4.02 ^dD^	3.96 ± 0.24 ^bB^	73.87 ± 1.26 ^dD^
Control	9.87 ± 0.71 ^aA^	—	15.16 ± 0.22 ^aA^	—

Values indicate the mean ± SD of three replicates. Different capital and small letters represent significant differences on 1% (*p* < 0.01) and 5% (*p* < 0.05) levels, respectively.

**Table 3 biomolecules-12-01304-t003:** The influence of pyraclostrobin and chitosan on *R. roxburghii* quality.

Treatments	Vitamin C (mg g^−1^)	Soluble Solid (%)	Soluble Protein (%)	Soluble Sugar (%)	Total Acidity (%)	Total Flavonoids (mg g^−1^)	Total Triterpenes (mg g^−1^)	SOD Activity (U g^−1^ FW)
P 100 + C 500	23.59 ± 0.55 ^a^	12.44 ± 0.32 ^a^	15.89 ± 0.51 ^a^	4.16 ± 0.18 ^a^	1.52 ± 0.06 ^a^	6.32 ± 0.31 ^a^	20.71 ± 0.35 ^a^	711.54 ± 21.38 ^a^
P 150	21.16 ± 0.69 ^b^	11.7 7± 0.74 ^ab^	14.65 ± 0.59 ^b^	3.83 ± 0.16 ^a^	1.41 ± 0.07 ^ab^	5.86 ± 0.23 ^b^	17.95 ± 0.65 ^bc^	668.15 ± 26.21 ^a^
P 100	19.73 ± 0.6 ^c^	10.89 ± 0.59 ^bc^	14.12 ± 0.72 ^bc^	3.19 ± 0.17 ^b^	1.36 ± 0.07 ^b^	5.32 ± 0.12 ^c^	17.20 ± 0.68 ^c^	608.93 ± 28.44 ^b^
C 500	21.05 ± 0.38 ^b^	11.85 ± 0.63 ^ab^	14.76 ± 0.63 ^b^	3.87 ± 0.19 ^a^	1.44 ± 0.09 ^ab^	5.93 ± 0.27 ^ab^	18.67 ± 0.55 ^b^	673.48 ± 17.68 ^a^
Control	18.04 ± 0.61 ^d^	10.53 ± 0.41 ^c^	13.41 ± 0.32 ^c^	3.08 ± 0.17 ^b^	1.23 ± 0.05 ^c^	5.16 ± 0.17 ^c^	15.16 ± 0.32 ^d^	556.89 ± 26.83 ^c^

Values indicate the mean ± SD of three replicates. Small letters represent significant differences on a 5% (*p* < 0.05) level.

**Table 4 biomolecules-12-01304-t004:** The influence of pyraclostrobin and chitosan on *R**. roxburghii* amino acids.

Treatments	EAA (mg kg^−1^)	NAA (mg kg^−1^)	TAA(mg kg^−1^)	The Percentage of EAA in TAA (%)	EAA/NAA
P 100 + C 500	88.47 ± 3.46 ^a^	339.22 ± 8.00 ^a^	461.76 ± 26.95 ^a^	19.17 ± 0.38 ^a^	0.26 ± 0.01 ^a^
P 150	75.29 ± 6.55 ^bc^	318.47 ± 10.41 ^abc^	414.28 ± 41.72 ^ab^	18.19 ± 0.26 ^b^	0.24 ± 0.01 ^ab^
P 100	68.86 ± 1.94 ^cd^	314.88 ± 14.11 ^bc^	401.33 ± 13.06 ^b^	17.16 ± 0.17 ^c^	0.22 ± 0.01 ^b^
C 500	78.46 ± 2.12 ^b^	321.95 ± 9.02 ^ab^	424.85 ± 12.33 ^ab^	18.47 ± 0.05 ^b^	0.24 ± 0.00 ^ab^
Control	64.82 ± 5.52 ^d^	296.68 ± 16.25 ^c^	381.84 ± 32.65 ^b^	16.98 ± 0.07 ^c^	0.22 ± 0.02 ^b^

Values indicate the mean ± SD of three replicates. Small letters represent significant differences on a 5% (*p* < 0.05) level.

## Data Availability

The datasets used or analyzed during the current study available from the corresponding author upon reasonable request.

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
