# Peer review of "Chitosan as an Adjuvant to Enhance the Control Efficacy of Low-Dosage Pyraclostrobin against Powdery Mildew of Rosa roxburghii and Improve Its Photosynthesis, Yield, and Quality"

_biomolecules, 2022, doi:10.3390/biom12091304_

Round 1

Reviewer 1 Report

In this study, the authors investigated the effects supplement of chitosan on the resistance on powdery mildew, photosynthesis, yield, quality and amino acids of Rosa roxburghii. The results showed that chitosan can be applied as an effective assist to promote pyraclostrobin against powdery mildew of R. roxburghii and improve its resistance, photosynthesis, yield, quality, and amino acids in the field. This manuscript is well-written and the research is interesting for readers.

Few comments:

1.     Misspellings in abstract, correct to “yield”.

2.     All figures are in the same pattern and color, please improved it.

3.     In Introduction, the authors just mentioned the occurrence of powdery mildew in Rosa roxburghii in Guizhou province of China. I would suggest the authors add more information on the disease occurrence and damages on the fruit production in the worldwide or other growing areas.

Author Response

1st Reviewer

Comment 1: In this study, the authors investigated the effects supplement of chitosan on the resistance on powdery mildew, photosynthesis, yield, quality and amino acids of Rosa roxburghii. The results showed that chitosan can be applied as an effective assist to promote pyraclostrobin against powdery mildew of R. roxburghii and improve its resistance, photosynthesis, yield, quality, and amino acids in the field. This manuscript is well-written and the research is interesting for readers.

Response: We sincerely thank the reviewer for the positive comments, careful reviews and warm work to our work! We also sincerely thank you for your hard corrections on our manuscript! Reviewers' comments are extremely constructive and valuable, and very helpful for revising and improving our manuscript, as well as the important guiding significance to our researches. We have studied carefully the reviewers' comments and made substantial revisions which we sincerely hope meet with approval. The responds to the reviewers' comments and the corrections in the revised manuscript are as flows. Thank you most sincerely!

Comment 2: 1. Misspellings in abstract, correct to “yield”.

Response: Thanks very much for the reviewer's careful reviews on our manuscript! I apologize for our carelessness! "yeild" has been revised as "yield" which marked in blue in the revised manuscript. Thank you most sincerely! (See lines 4, 14, 22, 25, 347)

Comment 3: 2. All figures are in the same pattern and color, please improved it.

Response: We sincerely thank the reviewer for the careful review and warm work to our manuscript! The pattern and color of all figures have been revised same in the revised manuscript. Thank you most sincerely! (See lines 211, 229, 248, 276)

Comment 4: 3. In Introduction, the authors just mentioned the occurrence of powdery mildew in Rosa roxburghii in Guizhou province of China. I would suggest the authors add more information on the disease occurrence and damages on the fruit production in the worldwide or other growing areas.

Response: We sincerely thank the reviewer for the careful reviews! The reviewers' comment is extremely valuable. The corresponding statement has been added and revised as "Recently, as an agricultural industry for boosting affluence and revitalizing rural, R. roxburghii industry has flourished rapidly in Guizhou Province in China with the planting areas of over 170,000 hm2, and has become the largest production area in the world [3,6]. Whereas, powdery mildew caused by Sphaerotheca sp. constantly occurs and seriously astricts the growth, yield, and quality of R. roxburghii, as well as frequently generates 30~40% of economic losses [1, 7]. As R. roxburghii is mainly produced in China, its powdery mildew also occurs frequently in Sichuan, Yunnan, Chongqing, southwestern Shaanxi, Hubei and Hunan production areas in China, equally often causing great economic losses [7, 8]." which marked in blue in the revised manuscript. Thank you most sincerely! (See lines 34-42, 423-424)

We appreciate for reviewer’s warm work earnestly, and hope that the correction will meet with approval. Once again, thank you very much for your comments and suggestions.

Reviewer 2 Report

This manuscript reports the efficacy of the application of chitosan biopolymer alone or in combination with a commercial fungicide pyraclostrobin in the control of powdery mildew disease of  Rosa roxburghii. The authors demonstrated that with the addition of chitosan 500 ppm, 50% synthetic fungicide use was reduced with high disease suppression. Furthermore, they showed that co-application of chitosan with fungicide also enhanced the yield, quality and amino acid contents in  Rosa roxburghii fruits. The findings of this study are generally interesting, however, the mechanistic aspects of the beneficial effects of the chitosan need to be discussed. As chitosan's effect on plants is elaborately been studied, based on the findings, a schematic diagram of the putative mechanisms of chitosan on growth, yield, and quality of fruits is needed for better understanding by the readers. The author should mention how they fix the dose of chitosan 500 ppm with the fungicide. Although quantitative data are presented, the author should add qualitative data (images of the results of in vivo experiments) along with the graphs. Overall, this manuscript may be accepted for publication after minor revision.

Author Response

2nd Reviewer

Comment 1: This manuscript reports the efficacy of the application of chitosan biopolymer alone or in combination with a commercial fungicide pyraclostrobin in the control of powdery mildew disease of  Rosa roxburghii. The authors demonstrated that with the addition of chitosan 500 ppm, 50% synthetic fungicide use was reduced with high disease suppression. Furthermore, they showed that co-application of chitosan with fungicide also enhanced the yield, quality and amino acid contents in  Rosa roxburghii fruits. The findings of this study are generally interesting, however, the mechanistic aspects of the beneficial effects of the chitosan need to be discussed. As chitosan's effect on plants is elaborately been studied, based on the findings, a schematic diagram of the putative mechanisms of chitosan on growth, yield, and quality of fruits is needed for better understanding by the readers. The author should mention how they fix the dose of chitosan 500 ppm with the fungicide. Although quantitative data are presented, the author should add qualitative data (images of the results of in vivo experiments) along with the graphs. Overall, this manuscript may be accepted for publication after minor revision.

Response: We sincerely thank the reviewer for the positive comments and warm work to our work! Reviewers' comments are extremely constructive and valuable, and very helpful for revising and improving our manuscript, as well as the important guiding significance to our researches. We have studied carefully the reviewers' comments and have substantial revisions which we sincerely hope meet with approval. The responds to the reviewers' comments and the corrections in the revised manuscript are as flows. Thank you most sincerely!

(1) As chitosan's effect on plants is elaborately been studied, based on the findings, a schematic diagram of the putative mechanisms of chitosan on growth, yield, and quality of fruits is needed for better understanding by the readers.

Response: Special thanks to you for your good comments and careful reviews! The putative mechanisms of chitosan-mediated plant growth regulation (Figure 1) was provided in the revised manuscript. Thank you most sincerely! (See lines 77-78, 99-100)

Figure 1. The putative mechanisms of chitosan-mediated plant growth regulation.

(2) The author should mention how they fix the dose of chitosan 500 ppm with the fungicide.

Response: Special thanks to you for your good comments! The corresponding statement has been added and revised as "For the preparation of P 100 + C 500, the appropriate amount of water was firstly used to dissolve 30% pyraclostrobin SC and chitosan respectively, then the 30% pyraclostrobin SC and chitosan diluents were mixed, and the required water was finally complemented." which marked in blue in the revised manuscript. Thank you most sincerely! (See lines 126-129)

(3) Although quantitative data are presented, the author should add qualitative data (images of the results of in vivo experiments) along with the graphs.

Response: Thanks very much for the reviewer's careful reviews and good comment on our manuscript! In fact, the results of in vivo experiments have been reported in the previous report of our research group. For reasons of scientific integrity, the qualitative data is not listed separately in the form of figures or tables in this study. The corresponding statement has been revised as "Subsequently, she further demonstrated that chitosan had a good toxicity against Sphaerotheca sp. with EC50 value of 416.21 mg kg−1, and the co-application of allicin and chitosan effectively controlled powdery mildew with the efficacy of 85.97%, as well as reliably enhanced the resistance, growth and quality of R. roxburghii [32]." and "In the previous report of our research group, Wu et al. [18] found that 30% pyraclostrobin SC could effectively control powdery mildew with the control efficacy of more than 90%, and notably promote its yield and quality." which marked in blue in the revised manuscript. We sincerely hope to get your understanding and recognition! Thank you most sincerely! (See lines 61-63, 80-84)

We appreciate for reviewer’s warm work earnestly, and hope that the correction will meet with approval. Once again, thank you very much for your comments and suggestions.